# Autotetraploid Emergence via Somatic Embryogenesis in *Vitis vinifera* Induces Marked Morphological Changes in Shoots, Mature Leaves, and Stomata

**DOI:** 10.3390/cells10061336

**Published:** 2021-05-28

**Authors:** Caterina Catalano, Loredana Abbate, Antonio Motisi, Dalila Crucitti, Vincenzo Cangelosi, Antonino Pisciotta, Rosario Di Lorenzo, Francesco Carimi, Angela Carra

**Affiliations:** 1Istituto di Bioscienze e BioRisorse, Consiglio Nazionale delle Ricerche, Corso Calatafimi 414, 90129 Palermo, Italy; caterina.catalano@ibbr.cnr.it (C.C.); loredana.abbate@ibbr.cnr.it (L.A.); antonio.motisi@ibbr.cnr.it (A.M.); dalila.crucitti@ibbr.cnr.it (D.C.); angela.carra@ibbr.cnr.it (A.C.); 2Dipartimento di Scienze Agrarie e Forestali, Università degli Studi di Palermo, Viale delle Scienze, Ed. 4, 90128 Palermo, Italy; vincenzocangelosi03@gmail.com (V.C.); antonino.pisciotta@unipa.it (A.P.); rosario.dilorenzo@unipa.it (R.D.L.)

**Keywords:** autopolyploidy, molecular analysis, ploidy variability, somatic embryogenesis, stomatal characteristics, grapevine

## Abstract

Polyploidy plays an important role in plant adaptation to biotic and abiotic stresses. Alterations of the ploidy in grapevine plants regenerated via somatic embryogenesis (SE) may provide a source of genetic variability useful for the improvement of agronomic characteristics of crops. In the grapevine, the SE induction process may cause ploidy changes without alterations in DNA profile. In the present research, tetraploid plants were observed for 9.3% of ‘Frappato’ grapevine somatic embryos regenerated in medium supplemented with the growth regulators β-naphthoxyacetic acid (10 µM) and N^6^-benzylaminopurine (4.4 µM). Autotetraploid plants regenerated via SE without detectable changes in the DNA profiles were transferred in field conditions to analyze the effect of polyploidization. Different ploidy levels induced several anatomical and morphological changes of the shoots and mature leaves. Alterations have been also observed in stomata. The length and width of stomata of tetraploid leaves were 39.9 and 18.6% higher than diploids, respectively. The chloroplast number per guard cell pair was higher (5.2%) in tetraploid leaves. On the contrary, the stomatal index was markedly decreased (12%) in tetraploid leaves. The observed morphological alterations might be useful traits for breeding of grapevine varieties in a changing environment.

## 1. Introduction

The possession of three or more complete sets of chromosomes is known as polyploidy and plays a crucial role in plant variation and evolution [1]. Whole genome duplication (WGD) is a major driver of adaptation, is responsible for angiosperm evolution [2], and is often a key factor in successful crop domestication [3]. Recent studies show physiological effects of WGD associated with increased dehydration stress tolerance in first-generation autotetraploids [4,5]. Moreover, a population genetic theory predicts both short- and long-term advantages of polyploidy [6]. Polyploidization is an important source of variability, with positive or negative outcomes [7], and is generally associated with increments of plant organs and alterations in stomatal density and size [8,9,10,11,12,13,14]. Polyploidization is also associated with an increase in tolerance and resistance to biotic [15] and abiotic stresses, such as salt and drought stresses [16,17,18,19,20,21,22].

Somatic embryogenesis (SE) is the induction of embryo generation from differentiated plant cells and it is an in vitro technique commonly used for clonal plant regeneration. The regeneration of somatic embryos in vitro has been used in a variety of applications such as virus elimination, cryopreservation, induced mutagenesis, and genetic engineering [23]. Tissue culturing is a source of stress for cells. Past studies revealed that regenerants obtained by in vitro cultures show genetic and epigenetic variations. This phenomenon was first described by Larkin and Scowcroft in 1981 [24], and was termed somaclonal variation. Therefore, in vitro clonal regeneration poses a problem of genetic stability [25]. On the other hand, it may represent an important source of novel variation. In some cases, plantlets regenerated in vitro through somatic embryogenesis may display chromosomal alterations and ploidy change, among other effects [26].

The grapevine (*Vitis vinifera* L.) is one of the most economically important woody crops cultivated worldwide, and more than 7000 cultivars are believed to exist around the world. So far, somatic embryos have been regenerated from different species of *Vitis* [23], and new plants bearing specific characteristics (virus-free, normally developed, and genetically true-to-type) have been obtained using a wide range of organs and tissues as explant source [27,28]. One of the most common starting tissues is anthers [29] but SE has also been obtained from ovaries, stigmas, and styles [30], anther filaments [31], and whole flowers [32]. Less commonly, tendrils, leaf disks, leaves, petioles, and stem nodal explants have been used for somatic embryogenesis induction [33]. In the grapevine, the SE induction process may cause DNA alterations [34] which can give rise to a large set of variants both in table and wine cultivars [35], providing a source of genetic variability useful for the improvement of agronomic characteristics of plants. A few cases in the literature report the occurrence of spontaneous ploidy change emergence via SE in the grapevine [36,37], however the effects of polyploidization on plants are not reported.

Different approaches are used to evaluate the genetic stability of grapevine plants regenerated from somatic embryos. Phenotypic identification of somaclonal variation can be evaluated based on the observation of morphological and physiological traits, using the descriptors from the Organization of Vine and Wine [38]. However, some changes obtained after in vitro culture cannot be observed *in planta*, because differences that influence the biological activity may not affect the phenotype [39]. The molecular basis of genetic variations such as transcription, transposed elements, chromosomal rearrangements, gene amplifications and gene mutations is now known [40] and molecular markers have become an important tool to check genetic uniformity. Among the DNA-based markers, random amplified polymorphic DNA (RAPD), inter simple sequence repeat (ISSR), and simple sequence repeat (SSR) have been used for the determination of genetic fidelity [41]. Flow cytometric analysis (FCM) is considered one of the most effective techniques to detect ploidy changes in plants regenerated in vitro and has been successfully applied to verify ploidy levels in several species. FCM does not require rapidly dividing cells, and sample preparation requires only a small amount of tissue [41,42,43]. This technique is based on the use of DNA-specific fluorochromes and on the analysis of the relative fluorescence of stained nuclei [44].

In the present study, the incidence of autotetraploid in somatic embryos of the grapevine was evaluated as polyploid breeding can be very useful for improving specific traits such as quality, yield, or environmental adaptation. In addition, we analyzed the polyploidization effects on several anatomical/morphological characters using 15 descriptors from the Organization of Vine and Wine (OIV) and six quantitative parameters regarding morphological characteristics of mature leaves.

## 2. Materials and Methods

### 2.1. Plant Material

Three varieties used for wine production were utilized in the present study, namely ‘Catarratto’, ‘Frappato’, and ‘Nero d’Avola’ [45,46]. The plant material was collected from the germplasm repository for perennial plants at the Institute of Biosciences and BioResources of the National Research Council of Italy (CNR-IBBR) located in Collesano district (province of Palermo), Italy (37°59′19.9″ N, 13°54′55.8″ E, 80 m a.s.l.). Different floral explants (pistil, anther/filament, and ovary) used for culture initiation were dissected from flowers harvested 15 days before anthesis.

### 2.2. Media and Culture Conditions and Plant Acclimatization

Explants were cultured on MS solid (6 g/L plant agar) medium [47] under three different plant growth regulator (PGR) combinations: (1) VV-4 medium, 5 µM N-(2-chloro-4-pyridyl)-N-phenylurea (4-CPPU) + 5 µM 2,4-dichlorophenoxy acetic acid (2,4-D); (2) VV-5 medium, 20 µM β-naphthoxyacetic acid (NOA) + 4 µM *N*-phenyl-*N*′-1,2,3-thiadiazol-5-ylurea (TDZ); and (3) VV-16 medium, 10 µM NOA + 4.4 µM N6-benzylaminopurine (BA) [48]. Media were supplemented with 88 mM sucrose. The pH of the media was adjusted to 5.7–5.8 with 1 N NaOH before autoclaving. All chemicals were purchased from Duchefa Biochemie, Netherland. Explants were incubated in a climatic chamber at 26 °C with a 16 h photoperiod (40 μmol m^−2^s^−1^ at shelf level, provided by Osram Cool White 18 W fluorescent lamps) and subcultured in the same culture medium at 60 d intervals. The explants showing embryogenic responses were transferred to basal MS-medium-deprived PGRs, supplemented with 88 mM sucrose and cultured for four more weeks to allow embryo proliferation and development. Then, germinated somatic embryos were collected and individually transferred to Magenta™ vessels containing 100 mL of basal solid MS medium under the same light and temperature conditions as described above, to allow further growth. After rooting, four plants produced from each somatic embryo (10–15 cm) were transferred to autoclaved Jiffy-7 peat pellets and moved onto a heating bench at 25 °C and high relative humidity (95–98%). After 4–5 weeks, plants were transferred into 2 L pots containing sterilized soil under natural daylight at 22/27 °C (night/day). After acclimatization, plantlets were transferred to the greenhouse located in Collesano district, grown in 20 L pots on a composite substrate of 2/3 peat and 1/3 agriperlite and fertilized with full strength nutrient solution as described by Oddo et al. [49]. After a 30–40-day period of acclimatization in the greenhouse the plants were transferred to field conditions for assessment of ampelographic traits.

### 2.3. Flow Cytometry Analysis

The DNA content was evaluated by FCM according to Carra et al. [50]. A total of 88 plantlets regenerated from somatic embryos were used and compared with the meristem tips of mother plants. The analysis was carried out with the Partec PAS flow cytometer (Partec GmbH, Munster, Germany), equipped with a mercury lamp. Fully expanded and young leaves (0.01 g) were chopped in a glass Petri dish with 1 mL nuclei extraction OTTO buffer 1 [51] and three drops of Tween 20. After 3 min, 1 mL of OTTO buffer 2 [51] was added. In addition, PVP-10 (1%) was added to plant samples to neutralize interference by cell metabolites in the analysis. The solution was filtered through a 30 μm Cell-Trics disposable filter Partec, and 400 µL of staining solution containing 4,6-diamidino-2-phenylindole (DAPI) was added. Routinely, 3000–4000 nuclei for each sample were measured, and histograms of DNA content were generated using the Partec software package (FloMax). Three replicates for each sample were carried out. The fluorescence intensity emitted was normalized by isolated nuclei from *Pisum sativum* L., optimal DNA reference standard for plant cytometric analyses [52]. This calibration was checked periodically.

### 2.4. Assessment of Genetic Stability in Regenerants by RAPD, ISSR, and SSR Markers

Regenerants coming from different embryogenic events, randomly chosen from three different explants cultured on different media, were compared to the mother plant for evaluation of genetic stability. Plant DNA was isolated from fresh leaves (100 mg) using the cetyltrimethylammonium bromide (CTAB) method as described by Doyle and Doyle [53]. The isolated genomic DNA was used for RAPD and ISSR analyses in order to assess mutations (Appendix A). The RAPD analysis of the grapevine genotypes was performed using eight random decamer primers—OPAT14, OPH15, OPM04, UBC219, UBC234, UBC239, UBC247, and UBC251 [54,55]. Seven ISSR primers—UBC834, UBC841, UBC848, UBC851, UBC855, ENEA7–9, and ENEA12 were used to amplify the genomic DNA [41,48,50]. For both molecular markers, polymerase chain reaction (PCR) amplification and analysis were performed as described by Carra et al. [48]. Only in those cases where the flow cytometry histograms revealed that the ploidy level of regenerated plants was different from that of the mother plant, an additional set of seven ISSR and nine simple sequence repeats (SSR) was tested for more accurate genetic stability evaluation. The additional ISSR analysis was carried out with ISSR2 + 2b ((AC)_8_YG; Ta 49 °C), ISSR 3 + 3b ((AG)_8_YC; Ta 49 °C), ISSR11 + 11b ((GA)_8_YC; Ta 56 °C), ISSR1-6 ((CA)_8_RG; Ta 49 °C), ENEA21 ((GA)_8_GG; Ta 49 °C), ENEA34 ((ACC)_6_CC; Ta 52 °C), and ENEA36 (CC(ATG)_6_; Ta 56 °C) [41,56,57]. SSR analysis was carried out using nine primer pairs—VVMD7, VVMD24, VVMD25, VVMD27, VVIb01, VVIh54, VVIp31, VVIp60, and VVIq52, distributed homogenously along the 19 chromosomes of diploid genome (475 Mbp and 2n = 38 chromosomes) of the grapevine (Appendix A). The SSR-PCR reactions were followed as described by De Michele et al. [58]; amplicons of each primer on all individuals were scored by an external service (Eurofins Genomics, Germany), and SSR allelic size was determined using the Gene Mapper v. 5.0 software. To confirm the reproducibility of the banding patterns, all analyses were repeated twice.

### 2.5. Ampelographic Analysis

Plants were phenotypically evaluated by morphological analysis of 15 descriptors from the Organization of Vine and Wine [38], relative to shoots and mature leaves (Table 1). Plant material for ampelographic analysis was collected in May (shoots) and July (leaves). The observations were performed on 15 fully expanded leaves (from the third node of several shoots) and 10 shoot tips for each embryogenic event.

### 2.6. Stomatal Characteristics

The effect of different ploidy levels on regenerants was also investigated by morphological analysis of mesophyll epidermal structures under light microscopy. Mature leaves were used to analyze six quantitative parameters regarding morphological characteristics of diploid and tetraploid regenerants as reported in Table 2. A few strips of epidermis were torn from the abaxial side of fully expanded leaves from diploid and tetraploid plants. Tissue segments were mounted on a microscope slide with a drop of distilled water and a coverslip to measure stomata size. The sizes of 20 stomata for each sample were evaluated under a light microscope (Optech Biostar BM 45 trinocular microscope with a Tucsen ISH 300 digital camera). Immediately after the preparation, the number of chloroplasts in the two guard cells of the same stomata was counted directly from microscope slide. Data were collected using ISCAPTURE Version 3.0. The stomatal index (SI) was also measured as (S/(E + S)) × 100, where S is the number of stomata and E is the number of epidermal cells per unit leaf area [59].

### 2.7. Statistical Analysis

Experiments were performed in a randomized complete block design with 10 replicates (Petri dishes) per treatment. Five explants each for stigma/style and ovary and 25 explants for anther/filament were used per plate. The embryogenic response of explants, the effect of different culture media, and the effect of explant type were expressed as a percentage on a Petri-dish basis and recorded 6 months after explant incubation. Percentages of embryo germination were recorded 2 months from the incubation of somatic embryos on PGR-free medium. The percentage data were arcsine square-root transformed prior to analysis. The results were back-transformed and presented as mean ± standard error. To highlight statistically significant differences and possible interactions between explant, medium, and genotype, the multi-way analysis of variance (ANOVA) was performed (*p* ≤ 0.05). One-way ANOVA was performed when the interaction between the factors was not significant. The separation of the averages was performed by Tukey’s test (*p* ≤ 0.05).

For the molecular analysis, only bands showing consistent amplification within the range of 200 bp to 3.5 kb were considered. Polymorphic ISSR and RAPD markers were scored for the presence (1) or absence (0) of bands for all the somaclones analyzed. All reactions were repeated at least twice, and only well-resolved, distinct, polymorphic, and reproducible bands across all runs were considered for analysis. Bands with the same migration were considered homologous fragments, independently of their intensity. Smeared DNA fragments and weak bands, which could not be readily distinguished, were excluded. In the SSR analysis, peak intensity was considered and compared with the internal size standard to estimate allele size as described by De Michele et al. [58]. For morphological evaluation, the data collected were subjected to statistical analysis using standard deviations of the mean and thereafter to ANOVA (*p* ≤ 0.05). Prior to analysis, percentage data were arcsine square-root transformed. Statistical analysis was performed using SigmaStat 3.5 for Windows.

## 3. Results

### 3.1. Somatic Embryogenesis

Explants produced a white callus 10–30 days after culture initiation, and somatic embryos emerged from callus surface 100 days after culture initiation. Figure 1 shows the embryogenic responses after six months of culture initiation in different treatments tested in three cultivars. The embryogenic potential varied greatly (0–22%) depending on explant type, genotype, and PGR combination used. The genotype was an important factor influencing plant regeneration. In fact, embryogenic capacity of the three cultivars (calculated across the media and explant types) ranged from 2.8% for ‘Nero d’Avola’ to 6.7% for ‘Frappato’ (Table 3). Regarding explant type (Table 4), pistils were more responsive (5.7%) compared to ovary (3.6%) and anther/filament (3.1%). Somatic embryogenesis occurred under all the culture conditions tested, with no significant differences according to the treatment tested (Table 4). Embryo germination percentages of ‘Frappato’ (55%), calculated across the medium and explant types, was lower than that of ‘Catarratto’ (63%) and ‘Nero d’Avola’ (67%). Plants maintained in vitro for 2 months after embryo germination reached an average height of 150 mm, and these well-developed plantlets could be transferred to Jiffy-7 peat pellets.

### 3.2. Ploidy Analysis

The ploidy level of regenerated plants was evaluated by FCM analysis by comparing nuclear DNA contents of the regenerants and the respective mother plant. A total of 31 regenerants of ‘Frappato’, 32 of ‘Catarratto’, and 25 of ‘Nero d’Avola’ were analyzed. The histograms revealed that the ploidy levels of most regenerated plants were the same as that of the mother plant (diploid, Figure 2A). However, different ploidy levels (tetraploid, Figure 2B) were detected in three (events 22, 33, and 34) ‘Frappato’ regenerated plants (9.3% of regenerants), all obtained in NOA (10 µM) and BA (4.4 µM) supplemented medium (VV-16) from pistil explants.

### 3.3. Assessment of Genetic Stability in Regenerants by RAPD, ISSR, and SSR Markers

To assess the genetic status of the progeny resulting from somatic embryogenesis, mother plants and regenerants were characterized by RAPD, ISSR, and SSR markers. The eight RAPD primers generated bands ranging from 200 to 3100 bp in size (Appendix A). The number of bands in the selected primers varied from 4 (UBC234) to 8 (OPAT14 and OPH15), with an average of 6.25 bands per RAPD primer. The 14 ISSR markers analyzed generated amplicons ranging from 300 to 3500 bp in size (Appendix A). The number of bands for each primer varied from 4 (ISSR2 + 2b) to 11 (ENEA7-9), with an average of approximately 7 bands per ISSR primer.

Both RAPD and ISSR analysis revealed homogeneity among regenerated plants and the in-vitro-obtained plants of the same cultivar (Figure 2C–D). No changes in the DNA profiles were detected by the additional set of seven ISSR showing monomorphic bands between tetraploid and diploid regenerants (Figure 3A,B).

Of nine SSR primer pairs used to compare the genetic stability between the three tetraploid plants of ‘Frappato’ and their mother plant (Appendix A), six SSRs revealed the presence of two alleles per locus in regenerants, as in the mother plant profiles (SSR profiles generated by marker VVMD27 are reported in Figure 3C). SSR profiles were also generated and the value of the size for each SSR locus were the same between the regenerants and the mother plant, confirming the genetic homogeneity through SSR marker-based DNA fingerprinting.

### 3.4. Ampelographic Analysis

To determine the influence of ploidy on morphological changes, diploid and tetraploid plants of ‘Frappato’ generated from different embryogenic events were analyzed. The vegetative behavior of diploid and tetraploid regenerants was assessed 4 years after acclimatization to overcome the juvenility phase. Phenotypes appeared homogeneous in all the diploid lines analyzed and therefore the diploid event N51 was chosen as a representative diploid plant for subsequent analyses. The plants derived from one of the tetraploid embryos did not survive and were therefore excluded from ampelographic analyses. To test the influence of the ploidy variation on morphological changes, a diploid (event N51) and two tetraploids (events N33 and N34, survived to ex vitro condition) regenerated plants were analyzed. The results of the ampelographic comparison between diploid and tetraploid events are reported in Table 1. The descriptions from the present study show that the diploid and tetraploid regenerants shared three (OIV 007, 008, and 067) out of 15 traits (in bold in Table 1), two of them regarding the shoot and one the mature leaf. The two tetraploid regenerants shared 11 out of 15 traits (underlined in Table 1), eight of them relative to mature leaf morphology and three to shoot. For example, the young shoot tip (OIV1) was fully open in tetraploid plants and closed in diploid ones (Table 1; Figure 4A–C). Also, the leaf morphology varied significantly between the tetraploid and diploid plants (Figure 4D–N). Only in four cases (asterisk in Table 1) did the two tetraploid forms differ from each other (OIV 065, 072, 076, and 078), yet for two traits (OIV 072 and 076) one of the two tetraploid plants was similar to the diploid (underlined in Table 1).

### 3.5. Leaf Morphological Characteristics and Stomata

The morphological characteristics of tetraploid grapevine leaves differed from those of the diploid counterparts (Table 2). Leaves of tetraploids were larger and longer than those of diploid ones. Compared with the diploid leaves, width and area of tetraploids were 31.9 and 87.5% higher, respectively (Table 2).

The stomata of tetraploid leaves were larger and longer that those of diploid ones (Table 2; Figure 4O–Q). The length and width of stomata of tetraploid leaves were 39.9 and 18.6% greater, respectively, than those of diploid leaves. The chloroplast number per guard cell pair was similar in tetraploid and diploid leaves. The SI was significantly lower in the two tetraploids (9.0 and 10.1) than in the diploid form (10.7) (Table 2).

## 4. Discussion

Polyploidization is a powerful strategy for genetic improvement and a successful method for inducing relevant physiological and morphological variations in plants [60]. Polyploidization events are associated with significant effects on plant anatomy and morphology. The volume of tetraploid cells is usually twice as high compared with that of diploid cells [61] and, as a consequence, the size of organs increases. Tetraploid plants often bear favorable horticultural traits, such as a larger fruits, sturdiness, high productivity, and tolerance to biotic and abiotic stresses [20,62]. Somatic embryogenesis is an efficient technique for clonal propagation; nevertheless, regenerated plants may exhibit somaclonal variations [24]. The rate of somaclonal variations is influenced by several stress factors such as wounding, exposure to sterilizing agents, plant growth regulators, sugar, or light conditions and can be particularly high when a callus phase or secondary embryogenesis is present [63]. The results reported in this paper show that grapevine plants regenerated via somatic embryogenesis may lead to somaclonal variations. Tetraploid plants have been obtained starting from floral explants cultivated in the presence of NOA and BA in one genotype, among the three genotypes analyzed. Several papers describe successful selections of somaclonal variants with a wide range of improved traits such as resistance to pests, diseases, and herbicides [64]. Somaclonal variation can also be considered as a tool for inducing ameliorative variations, as reported for several crops such as hazel nut [65], ray [66], *Citrus paradisi* [67], oil palm [68], potato [69], coffee [70], olive [71], and sugarcane [72]. In the grapevine, somaclonal variation can be spontaneous or induced [73]. Some mutated traits have been described in plants regenerated from nucellar tissue culture in vitro, such as reduction in the size of the berries, a decrease in yield of about 50%, and a marked increase in the sugar content [74]. Mutated individuals obtained with our experimental procedure showed morphological differences in shoots, mature leaf shape, and stomata characteristics. Ampelographic analysis showed markedly different phenotypes between diploid and tetraploid forms, with 12 out of 15 traits diverging. Conversely, comparison of OIV descriptors showed a similar phenotype in all the tetraploid regenerants, which shared 80% of traits, with very minor discrepancies. Their similarity is also confirmed by the biometric analysis of the mature leaves, which were significantly larger in the two tetraploid lines than in diploids. The variability in leaf morphology, i.e., the degree of opening of petiole and lateral sinuses, the number of lobes, the undulations or blistering of the blade, and the goffering and the profile of the blade in a cross section, may influence the mechanisms of the responses of plants to external stimuli, such as disease resistance [75]. We observed that stomata dimension and density are influenced by polyploidization, being larger in size and less numerous in tetraploids. This characteristic can affect infections, as stomata are one of the major portals for pathogen penetration [76]. Studies carried out on several grapevine varieties reported a positive correlation between susceptibility to downy mildew and stomatal density [75].

A callus phase during the initial stages of the regeneration pathway can facilitate the onset of chromosomal alterations and changes in the ploidy level [77]. Since in our culture conditions, during culture initiation, a callus phase was detected, the genetic stability of regenerants was verified. Somaclonal variants can be identified using different techniques both for morphological and molecular traits. Flow cytometry has been used in the grapevine to verify the ploidy level and ploidy stability of somatic embryogenesis-derived plants [36,39]; flow cytometry can easily and accurately distinguish tetraploid from diploid plants. However, in the grapevine the high number and the reduced dimension of the chromosomes (2n = 38) make karyotyping an inconclusive strategy. Moreover, cytosolic phenolics, such as tannic acid, negatively influence flow cytometry results [39]. Our procedure presents several ameliorative aspects: a detergent and PVP-10 have been added to facilitate the release of the nuclei and to obtain an ideal nuclear suspension; the sampling of young leaves gives an optimal number of intact nuclei, resulting in better fluorescence; and frequently replacing the scalpel blade reduces mechanical damage to the nuclei. Therefore, the protocol developed in this research can be used as a reference for ploidy and stability analysis in *V. vinifera.*

Molecular techniques, such as RAPD and microsatellite or ISSR, are valuable tools for analyzing the genetic fidelity of plants regenerated in vitro. Genetic profiles determined by DNA markers have shown that genetic fidelity is not compromised during somatic embryogenesis, except in very rare instances [39,78]. In our experimental system, even if a callus phase was observed during the incubation of explants, analysis with molecular markers (ISSR, SSR, and RAPD) indicated no differences between regenerants and mother plants, thus confirming previous studies on other species regenerated from flower-dissected explants [50,79].

Since molecular marker profiles were monomorphic in all regenerants, we hypothesize that there was not a significant genome rearrangement after whole genome duplication. However, we observed that different ploidy levels induced several morphological changes of the shoots, mature leaves, and stomata in tetraploid regenerants. Alterations in leaf morphology, such as those present in our tetraploid regenerants, may enhance resistance to fungal pathogens. The stability of such changes along generations will be tested and favorable traits incorporated into an ongoing grape-breeding program at our institute.

## 5. Conclusions

This manuscript addresses the topic of alterations of the ploidy in grapevines regenerated in vitro. Nine percent of regenerants were tetraploid and showed profound anatomical/morphological changes in shoots, mature leaves, and stomata.

## Figures and Tables

**Figure 1 cells-10-01336-f001:**
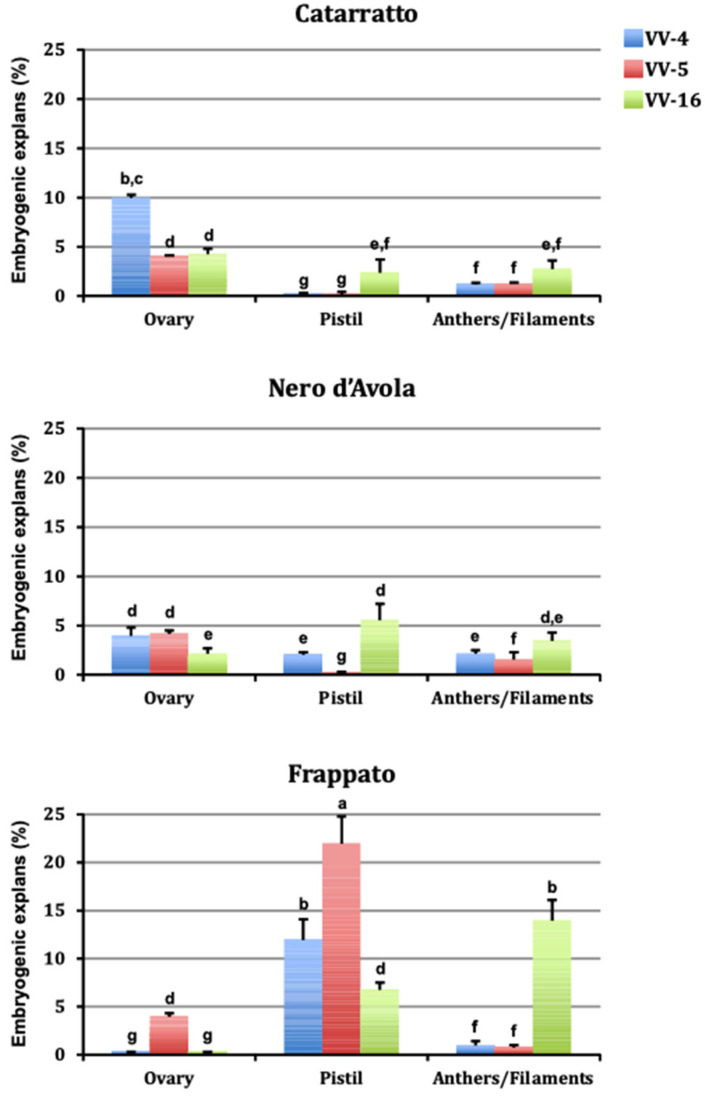
Percentages of embryogenic explants of ‘Catarratto’, ‘Nero d’Avola’, and ‘Frappato’ using three explant types and three PGR combinations. Data were collected 6 months after culture initiation and each treatment comprised 250 explants for anthers/filaments and 50 explants for ovaries and pistils. Means ± SE, values followed by the same letter are not significantly different at *p* < 0.05 level (Tukey’s test).

**Figure 2 cells-10-01336-f002:**
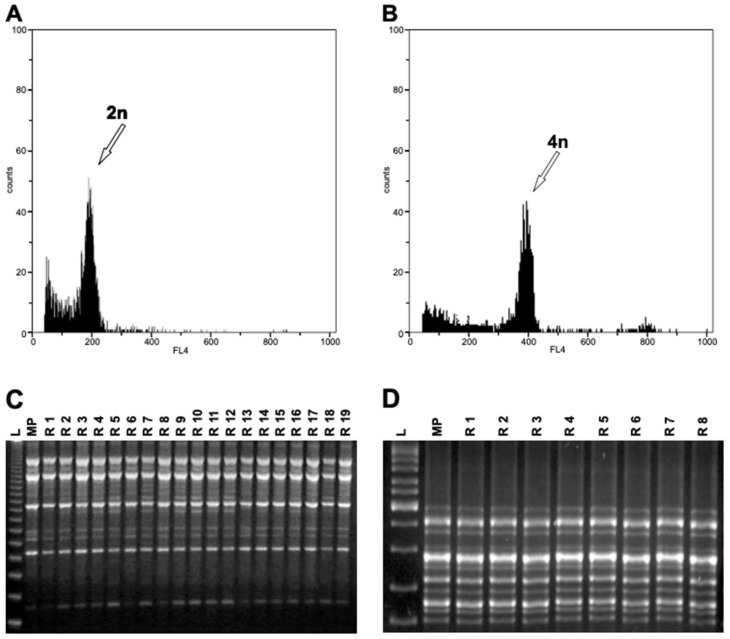
Ploidy level and genetic stability of regenerants through flow cytometric analysis and ISSR markers. Representative flow cytometric histograms of nuclei isolated from leaves of diploid ((**A**) = event 51) and tetraploid plants ((**B**) = event 34). Histograms showed the fluorescence intensity of diploids on channel 200 and that of tetraploids on channel 400. The data were obtained with the same instrument setting. The gain was positioned on channel 200 for diploid nuclei, while tetraploid nuclei peak appeared on channel 400. (**C**) DNA profiles of ‘Frappato’ regenerants amplified with the RAPD primer OPAT-14. L, ladder 123-bp; MP, mother plant; R 1–19, in vitro regenerants. (**D**) Genetic fidelity assessment of ‘Frappato’ regenerants with ISSR marker (ENEA7-9). L, ladder Thermo Scientific™ GeneRuler™ DNA Ladder Mix; MP, mother plant; R 1–8, in vitro regenerants. Amplification profiles were monomorphic across all regenerants.

**Figure 3 cells-10-01336-f003:**
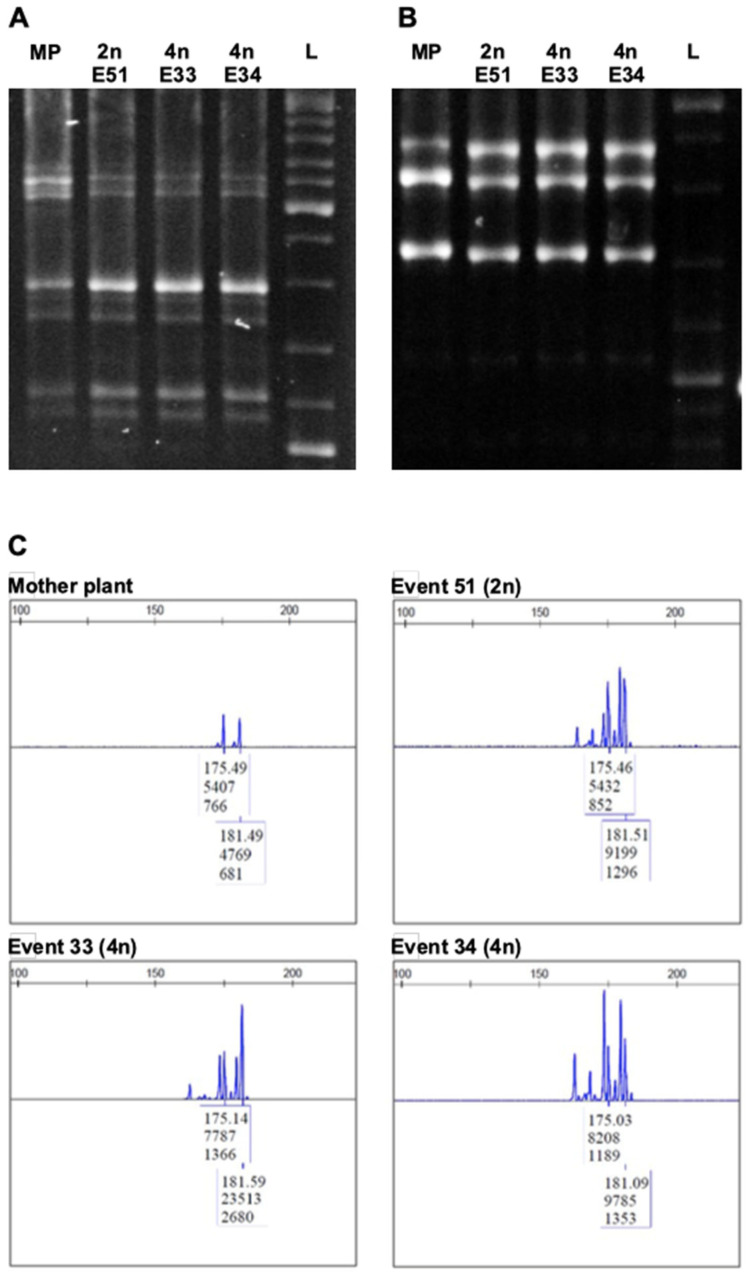
Genetic fidelity assessment of diploid (event 51) and tetraploid (events 33 and 34) ‘Frappato’ regenerants with ISSR markers. Profiles were obtained with: (**A**) Primer ISSR1-6; L, ladder Thermo Scientific™ GeneRuler™ DNA Ladder Mix; MP, mother plant; E51, event 51 (2n); E33, event 33 (4n); E34, event 34 (4n). (**B**) Primer ISSR2 + 2b. L, ladder Thermo Scientific™ GeneRuler™ DNA Ladder Mix; MP, mother plant; E51, event 51 (2n); E33, event 33 (4n); E34, event 34 (4n). Amplification products were monomorphic across all regenerants. (**C**) SSR profiles generated by SSR marker VVMD27 of ‘Frappato’ mother plant, event 51 (2n), event 33 (4n), and event 34 (4n).

**Figure 4 cells-10-01336-f004:**
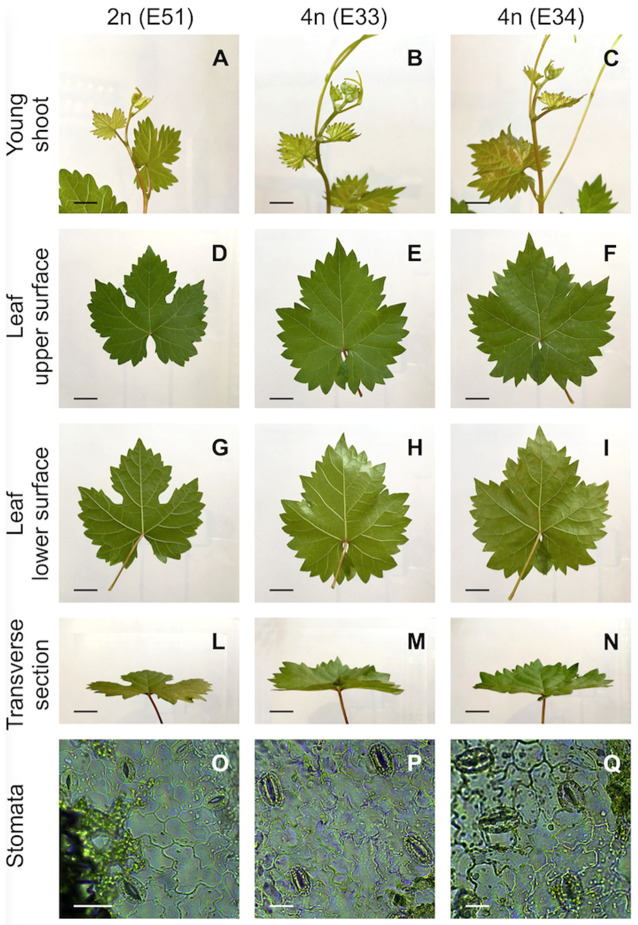
Morphological changes of the shoots, mature leaves, and stomata of diploid (event 51) and tetraploid (events 33 and 34) regenerants. (**A**–**C**) Young shoot (bar = 1.5 cm); (**D**–**F**) mature leaves upper surface (bar = 2 cm); (**G**–**I**) mature leaves lower surface (bar = 2 cm); (**L**–**N**) mature leaves transverse section (bar = 2 cm); and (**O**–**Q**) stomata in mature leaves (bar = 20 μm).

**Table 1 cells-10-01336-t001:** Ampelographic characteristics of 2n and 4n regenerants based on the 15 OIV descriptors used (OIV, 2009).

OIV Code	OIV Descriptor	Levels	2n (E51)	4n (E33)	4n (E34)
OIV 001	Young shoot: opening of the shoot tip	(1) Closed; (3) half open; (5) fully open	1	5	5
OIV 007	Shoot: color of the dorsal side of internodes	(1) Green; (2) green and red; (3) red	**2**	**2**	**2**
OIV 008	Shoot: color of the ventral side of internodes	(1) Green; (2) green and red; (3) red	**2**	**2**	**2**
OIV 065	Mature leaf: size of blade	(1) Very small; (3) small; (5) medium; (7) large; (9) very large	3	5 *	7 *
OIV 067	Mature leaf: shape of blade	(1) Cordate; (2) wedge-shaped; (3) pentagonal; (4) circular; (5) kidney-shaped	**3**	**3**	**3**
OIV 068	Mature leaf: number of lobes	(1) One (entire leaf); (2) three; (3) five; (4) seven; (5) more than seven	3	2	2
OIV 072	Mature leaf: goffering of blade	(1) Absent or very weak; (3) weak; (5) medium; (7) strong; (9) very strong	1	1 *	3 *
OIV 074	Mature leaf: profile of blade in cross section	(1) Flat; (2) V-shaped; (3) involute; 4) revolute; (5) twisted	1	2	2
OIV 075	Mature leaf: blistering of upper side of blade	(1) Absent or very weak; (3) weak; (5) medium; (7) strong; (9) very strong	1	3	3
OIV 076	Mature leaf: shape of teeth	(1) Both sides concave; (2) both sides straight; (3) both sides convex; (4) one side concave, one side convex; (5) mixture between, both sides straight and both sides convex	4	5 *	4 *
OIV 078	Mature leaf: length of teeth compared with their width	(1) Very short; (3) short; (5) medium; (7) long; (9) very long	3	5 *	7 *
OIV 079	Mature leaf: degree of opening/overlapping of petiole sinuses	(1) Very wide open; (3) open; (5) closed; (7) overlapped; (9) strongly overlapped	3	7	7
OIV 082	Mature leaf: degree of opening/overlapping of upper lateral sinuses	(1) Open; (2) closed; (3) slightly overlapped; (4) strongly overlapped; (5) absence of sinus	1	5	5
OIV 093	Mature leaf: length of petiole compared to length of middle vein	(1) Much shorter; (3) slightly shorter; (5) equal; (7) slightly longer; (9) much longer	3 (0.72)	1 (0.56)	1 (0.59)
OIV 094	Mature leaf: depth of upper lateral sinuses	(1) Absent or very shallow; (3) shallow; (5) medium; (7) deep; (9) very deep	5	1	1

Identical characteristics among all samples are in bold. The asterisk (*) indicates the values different between the two tetraploid plants. The underlined values refer to identical characteristics observed between a diploid and a tetraploid, which is in turn different from the other tetraploid.

**Table 2 cells-10-01336-t002:** Morphological comparison of diploid and tetraploid leaves.

Characteristics	2n (E51)	4n (E33)	4n (E34)
Leaf length (cm)	8.4 ± 0.7 ^a^	9.5 ± 1 ^a^	9.5 ± 0.5 ^a^
Leaf width (cm)	9.4 ± 1.1 ^a^	13 ± 0.5 ^b^	11.8 ± 0.3 ^b^
Leaf area (cm^2^)	43.3 ± 8.7 ^a^	79.8 ± 6.1 ^b^	82.6 ± 3.1 ^b^
Stomatal length (μm)	18.3 ± 0.5 ^a^	27.1 ± 0.5 ^b^	24.1 ± 0.5 ^b^
Stomatal width (μm)	13.4 ± 0.3 ^a^	16.3 ± 0.3 ^b^	15.5 ± 0.6 ^b^
Number of chloroplastsper guard cell pair	44.3 ± 2.8 ^a^	47.1 ± 2.6 ^a^	46.1 ± 2.6 ^a^
SI	10.7 ± 0.3 ^a^	10.1 ± 1.1 ^b^	9.0 ± 0.5 ^b^

Stomatal index (SI) values are also reported. Data represent mean ± standard deviation (*n* = 10). Different lowercase letters within rows indicate a significant difference at the 5% level (*p* < 0.05).

**Table 3 cells-10-01336-t003:** Genotype specificity of somatic embryogenesis.

Genotype	Embryogenic Explants (%)
‘Catarratto’	2.9 ± 0.2 ^b^
‘Nero d’Avola’	2.8 ± 0.6 ^b^
‘Frappato’	6.7 ± 0.7 ^a^

Somatic embryogenesis data were collected 6 months after culture initiation. Means + SE, values followed by the same letter are not significantly different at *p* < 0.05 level (Tukey’s test).

**Table 4 cells-10-01336-t004:** Effect of three PGR combinations (VV-4, VV-5, and VV-16) and three explant types (ovary, pistil, and anther/filament) on somatic embryogenesis.

Media	Embryogenic Explants (%)	Explant Type	Embryogenic Explants (%)
VV-4 (5 µM CPPU + 5 µM 2,4-D)	3.6 + 0.2 ^a^	Ovary	3.6 + 0.2 ^b^
VV-5 (20 µM NOA + 4 µM TDZ)	4.2 + 0.3 ^a^	Pistil	5.7 + 0.6 ^a^
VV-16 (10 µM NOA + 4.4 µM BA)	4.6 + 0.1 ^a^	Anther/filament	3.1 + 0.2 ^b^

Data were collected 6 months after culture initiation. Means + SE, in each column’s values followed from the same letter are not significantly different at *p* < 0.05 level (Tukey’s test).

## Data Availability

Not applicable.

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
