# Peer review of "Autotetraploid Emergence via Somatic Embryogenesis in *Vitis vinifera* Induces Marked Morphological Changes in Shoots, Mature Leaves, and Stomata"

_cells, 2021, doi:10.3390/cells10061336_

Round 1

Reviewer 1 Report

Although I overall agree with the position of the authors concerning polyploidy, I feel that throughout the manuscript (namely the Introduction and Discussion) the concept is mixing outcomes occurring in natural species (polyploids) with the domestication of crops. Truly, whole genome duplication is prevalent in angiosperms and polyploids and might have an evolutionary advantage, but that does not occur in most of crops. In fact, an outcome of the domestication process is that genetic diversity of crops is now known to be quite limited, and for that scientists are going back to crop wild relatives. Despite that, some “recent” crop polyploids show some advantages under stress conditions. See for instance:

https://doi.org/10.1105/tpc.19.00397

https://doi.org/10.1038/s41438-020-0311-7

https://doi.org/10.3390/ijms22063125

I do agree with sentences as “polyploids may act as ‘sponges’ accumulating adaptive allelic diversity” but not in the context of this paper. The former sentence is true but in the context of natural allopolyploids, which is not the case here. As the authors have reported autopolyploids show little (or none) genetic divergence regarding parental lines. So, I would suggest re-writing some of the sentences to avoid misleading conclusions.

Please define the aims more clearly. The authors state “In the present study, the spontaneous incidence of autotetraploid in somatic embryos of grapevine was evaluated.” – For what?

“In addition, we analysed the polyploidization effects on several anatomical/morphological characters in autotetraploid plants regenerated from somatic embryos” – I would suggest referring that the use of 15 vine descriptors. In addition, why did you used only qualitative traits?

Please give more details in the statistical analysis. “Effects of PGR treatment, genotype and type of explant were tested by ANOVA (P ≤ 0.05), and the differences among means were tested by Tukey’s test.” Differences among means of what? Among or between? What about the interactions of these effects?

The use of RAPDs is not particular useful. By now I think it has been widely shown that this type of markers is not reproducible. So, I am surprised to see the authors pointing it as a valuable tool.

Author Response

We are most grateful to the reviewer for such an attentive reading of our article and valuable comments. We have done our best to follow all recommendations of the reviewer.

  1. Reviewer’s remark: Although I overall agree with the position of the authors concerning polyploidy, I feel that throughout the manuscript (namely the Introduction and Discussion) the concept is mixing outcomes occurring in natural species (polyploids) with the domestication of crops. So, I would suggest re-writing some of the sentences to avoid misleading conclusions.
  2. Reviewer’s remark: I do agree with sentences as “polyploids may act as ‘sponges’ accumulating adaptive allelic diversity” but not in the context of this paper. The former sentence is true but in the context of natural allopolyploids, which is not the case here. As the authors have reported autopolyploids show little (or none) genetic divergence regarding parental lines. So, I would suggest re-writing some of the sentences to avoid misleading conclusions.

Authors Response: We agree with the remark and we changed the sentence eliminating the “and new experimental data support the hypothesis that polyploids may act as ‘sponges’ accumulating adaptive allelic diversity [6].” Moreover, we added 2 new references about polyploidization and stress resistance. The new references are:

- Wei, T., Wang, Y., & Liu, J. H. (2020). Comparative transcriptome analysis reveals synergistic and disparate defense pathways in the leaves and roots of trifoliate orange (Poncirus trifoliata) autotetraploids with enhanced salt tolerance. Horticulture Research, 7(1), 1-14. https://doi.org/10.1038/s41438-020-0311-7;

- Marques, I., Fernandes, I., Paulo, O. S., Lidon, F. C., DaMatta, F. M., Ramalho, J. C., & Ribeiro-Barros, A. I. (2021). A Transcriptomic Approach to Understanding the Combined Impacts of Supra-Optimal Temperatures and CO2 Revealed Different Responses in the Polyploid Coffea arabica and Its Diploid Progenitor C. canephora. International journal of molecular sciences, 22(6), 3125. https://doi.org/10.3390/ijms22063125.

  1. Reviewer’s remark: Please define the aims more clearly. The authors state “In the present study, the spontaneous incidence of autotetraploid in somatic embryos of grapevine was evaluated.” – For what?

Authors Response: The sentence was changed as reported: “In the present study, the incidence of autotetraploid in somatic embryos of grapevine was evaluated as polyploid breeding can be very useful for improving specific traits such as quality or environmental adaptation.”

  1. Reviewer’s remark: “In addition, we analysed the polyploidization effects on several anatomical/morphological characters in autotetraploid plants regenerated from somatic embryos” – I would suggest referring that the use of 15 vine descriptors. In addition, why did you used only qualitative traits?

Authors Response: To analyze the polyploidization effects we used 15 qualitative vine parameters. Nevertheless, among the 15 parameters used, one was both qualitative and quantitative. Parameter 093 gives the ratio between length of petiole compared to length of middle vein. Moreover, in table 4 are reported data on 6 quantitative parameters that give measurements of fully expanded mature leaves. We decided not to use quantitative OIV traits on fruits because, even if plants are 4 years old, they are not in full production and data about berry, must and total yield variables may not be significant.

The text was improved in section 2.6. Stomatal characteristics as reported: “Mature leaves were used to analyse six quantitative parameters regarding morphological characteristics of diploid and tetraploid regenerants as reported in table 4.”

  1. Reviewer’s remark: Please give more details in the statistical analysis. “Effects of PGR treatment, genotype and type of explant were tested by ANOVA (P ≤ 0.05), and the differences among means were tested by Tukey’s test.” Differences among means of what? Among or between? What about the interactions of these effects?

Authors Response: Data represent differences among percentage means obtained scoring embryogenic events on the base of Petri dishes. A Petri dish is considered a replicate. The “statistical analyses” section was improved as reported: “Experiments were performed in a randomised complete block design with 10 replicates (Petri dishes) per treatment. Five explants each for stigma/style and ovary and 25 explants for anther/filament were used per plate. Embryogenic response of explants, effect of different culture media and effect of explant type were expressed as percentage on a Petri dish basis and recorded 6 months after explant incubation. Percentages of embryo germination were recorded 2 months from the incubation of somatic embryos on PGR-free medium. The percentage data were arcsin square-root transformed prior to analysis. The results were back transformed and presented as mean± standard error. To highlight statistically significant differences and possible interactions between explant, medium and genotype, the multi-way analysis of variance (ANOVA) was performed (p ≤ 0.05). One-way ANOVA was performed when the interaction between two factors was not significant. The separation of the averages was performed by Tukey’s test (p ≤ 0.05).”

Interaction between variables was not reported because data were not statistically significative as explained in Statistical analyses section.

  1. Reviewer’s remark: The use of RAPDs is not particular useful. By now I think it has been widely shown that this type of markers is not reproducible. So, I am surprised to see the authors pointing it as a valuable tool.

Authors Response: Several authors showed a careful optimization of the amplification reaction to achieve the satisfactory reproducibility of the RAPD data. Please find herewith some references:

- Emilia Jabłońska (2020) Molecular diversity of the Fusarium fujikuroi species complex from maize. European Journal of Plant Pathology, 158 (4) : 859, DOI: 10.1007/s10658-020-02121-7;

- Walter P. Pfliegler (2014) Diversity of Candida zemplinina isolates inferred from RAPD, micro/minisatellite and physiological analysis. Microbiological Research, 169 (5-6) : 402, DOI: 10.1016/j.micres.2013.09.006;

- M. Skoric, B. Siler, T. Banjanac, J. Zivkovic, S. Dmitrovic, D. Misic, D. Grubisic (2012) The reproducibility of RAPD profiles: effects of PCR components on RAPD analysis of four Centaurium species. Arch Biol Sci, 64, pp. 191-199, DOI: 10.2298/ABS1201191S;

- Padmalatha, K. and M. N. V. Prasad (2006) Optimization of DNA isolation and PCR protocol for RAPD analysis of selected medicinal and aromatic plants of conservation concern from Peninsular India. Afr. J. Biotechnol. 5, 230-234, DOI: 10.5897/AJB05.188.

The PCR conditions for RAPD analysis can be optimized by varying the concentrations of the reaction mixture components (type and concentration of thermostable polymerase, deoxynucleotide triphosphates, Mg2+ ions, primer and DNA template concentration), and other reaction factors such as primer annealing, primer extension, denaturation time and temperature.

Furthermore, several recent papers reported the use of RAPD as molecular tools to investigate both in plants and in other organisms.

Please find some examples of recently published papers about use of RAPD in plants:

- Amom et al. (2020) Efficiency of RAPD, ISSR, iPBS, SCoT and phytochemical markers in the genetic relationship study of five native and economical important bamboos of North-East India. PHYTOCHEMISTRY, DOI: 10.1016/j.phytochem.2020.112330 (IF: 3.044);

- Sahay et al. (2020) Photosynthetic activity and RAPD profile of polyethylene glycol treated B. juncea L. under nitric oxide and abscisic acid application. JOURNAL OF BIOTECHNOLOGY, DOI: 10.1016/j.jbiotec.2020.03.004 (IF: 3.503);

- Narvaez et al. (2019) Plant Regeneration via Somatic Embryogenesis in Mature Wild Olive Genotypes Resistant to the Defoliating Pathotype of Verticillium dahliae. FRONTIERS IN PLANT SCIENCE, DOI: 10.3389/fpls.2019.01471 (IF: 4.402);

- Venkatachalam et al. (2017) Zinc oxide nanoparticles (ZnONPs) alleviate heavy metal-induced toxicity in Leucaena leucocephala seedlings: A physiochemical analysis. PLANT PHYSIOLOGY AND BIOCHEMISTRY, DOI: 10.1016/j.plaphy.2016.08.022 (IF:3.72);

- Kim et al. (2021) Genetic Diversity of Castor Bean (Ricinus communis L.) Revealed by ISSR and RAPD Markers. AGRONOMY-BASEL, DOI: 10.3390/agronomy11030457 (IF: 2.603);

- Han et al. (2017) The Evolution of Vicia ramuliflora (Fabaceae) at Tetraploid and Diploid Levels Revealed with FISH and RAPD. PLOS ONE, DOI: 10.1371/journal.pone.0170695 (IF: 2.74);

- Sirijan et al. (2020) Characterisation of Thai strawberry (Fragaria x ananassa Duch.) cultivars with RAPD markers and metabolite profiling techniques. PHYTOCHEMISTRY, DOI: 10.1016/j.phytochem.2020.112522 (IF: 3.044);

- Islam et al. (2019) Agro-Morphological, Yield, and Genotyping-by-Sequencing Data of Selected Wheat (Triticum aestivum) Germplasm From Pakistan. FRONTIERS IN GENETICS, DOI: 10.3389/fgene.2021.617772 (IF: 3.26);

- Pan et al. (2017) Development of an RAPD-based SCAR marker linked with smut disease resistance in commercial sugarcane cultivars. PHYTOPATHOLOGY (IF: 3.234);

- Zhou et al. (2018) Molecular Sex Identification in Dioecious Hippophae rhamnoides L. via RAPD and SCAR Markers. MOLECULES, DOI: 10.3390/molecules23051048 (IF: 3.267);

- Raza et al. (2020) Polymorphic information and genetic diversity in Brassica species revealed by RAPD markers. BIOCELL, DOI: 10.32604/biocell.2020.010207 (IF: 2.821);

- Honore et al. (2020) Effects of the size of papaya (Carica papaya L.) seedling with early determination of sex on the yield and the quality in a greenhouse cultivation in continental Europe. SCIENTIA HORTICULTURAE, DOI: 10.1016/j.scienta.2020.109218 (IF: 2.769);

Here some recently published papers using RAPD in other organisms are listed:

- Liu et al. (2021) The exposure of gadolinium at environmental relevant levels induced genotoxic effects in Arabidopsis thaliana (L.). ECOTOXICOLOGY AND ENVIRONMENTAL SAFETY, DOI: 10.1016/j.ecoenv.2021.112138 (IF: 4.872);

- Nafarrate et al. (2021) Isolation, host specificity and genetic characterization of Campylobacter specific bacteriophages from poultry and swine sources. FOOD MICROBIOLOGY, DOI: 10.1016/j.fm.2021.103742 (IF: 4.155).

Reviewer 2 Report

This manuscript describes the appearance of tetraploid individuals following somatic embryogenesis in grapevine. The data demonstrating this polyploidy and the associated phenotypic alterations are convincing.

However, there is an assertion in the abstract and in the body of the manuscript that these polyploid individuals are without detectable changes in their DNA profiles. There are two problems with this statement. Firstly, only a small number of markers have been used to demonstrate genomic integrity. It is well known that somaclonal variants usually have genomic variations limited to a subset of regions of the genome. There is no indication if any of the markers used are variant when testing 2n somaclonal off types indicating that they are useful markers for determining genomic integrity through the tissue culture process. Secondly, the data in Figure 3 C appears to demonstrate that this marker is one that can detect genomic variation resulting from the tissue culture process. The profiles of the three regenerants are clearly different from the parent and also from each other. Therefore, the statement that the regenerants are without detectable changes in the DNA profiles is not supported by the data.

Author Response

We are most grateful to the reviewer for critical comments, which helped us to improve the manuscript.

  1. Reviewer’s remark: It is well known that somaclonal variants usually have genomic variations limited to a subset of regions of the genome. There is no indication if any of the markers used are variant when testing 2n somaclonal off types indicating that they are useful markers for determining genomic integrity through the tissue culture process.

Authors Response: In regenerated plants chromosomal rearrangements are an important source of variation. Typical genetic alterations are: changes in chromosome numbers (polyploidy and aneuploidy), chromosome structure (translocations, deletions, insertions, and duplications), and DNA sequence (base mutations). In general, the use of one type molecular marker to assess the stability of in vitro propagated plants may be insufficient. Several authors used multiple molecular marker types to study somaclonal variation in regenerants of several plant species. ISSRs are randomly distributed throughout the genome and fragments from multiple loci are generated simultaneously, allowing to obtain multilocus fingerprinting profiles.

  1. Reviewer’s remark: Secondly, the data in Figure 3 C appears to demonstrate that this marker is one that can detect genomic variation resulting from the tissue culture process. The profiles of the three regenerants are clearly different from the parent and also from each other. Therefore, the statement that the regenerants are without detectable changes in the DNA profiles is not supported by the data.

Authors Response: During the PCR amplification of di‐, tri‐, and tetranucleotide microsatellite loci, artifacts are produced and can complicate data interpretation. Commonly, PCR artifact(s) may be created during amplification, incomplete terminal adenylation (“Plus A additions”), mono or dinucleotide stutter, or pull-up due to spectral overlap of two fluorophores. In our case, Figure 3 shows the multipeak pattern with stutter peaks from heterozygous individuals. Stutter artifacts are observed as multiple peaks preceding the true allele peak. Stutter peaks may be caused by polymerase slippage during elongation. The number of peaks and their intensities are proportional to the length of the repeat and the number of repeats in the PCR product. Shorter repeat units (di‐ or tri‐, for ex.) generate more stutters, and dinucleotide repeats tend to generate more stutter peaks than trinucleotide repeats. Applied Biosystems GeneMapper® software is designed to identify and filter out stutter artifacts for accurate scoring (peak height and size) of true alleles. In Figure 3 the size of our fragments is reported and it corresponds to 175 bp and 181 bp for all four samples analyzed. This confirm that no genomic variation is observed among the genetic profiles of the three regenerants and among them and the genetic profile of the mother plant.

Round 2

Reviewer 1 Report

Thank you for the time and efforts trying to solve my comments. I think the authors have solved them all. Congratulations.